# Differential epigenetic regulation of glucose-induced alteration of miR-9 in retinal and cardiac endothelial cells

Eric Wang, Biao Feng, Shali Chen, Subrata Chakrabarti*

Department of Pathology and Laboratory Medicine, Western University, London, Ontario, Canada

* subrata.chakrabarti@schulich.uwo.ca

## Abstract

Diabetic complications are significant causes of reduced quality of life among people living with diabetes. Endothelial dysfunction is a core component of diabetic complications. Microvascular endothelial cells from the heart and retina, two major targets of microvascular complications of diabetes, respond largely uniquely to diabetic insult. However, both involve inhibition of microRNA 9 (miR-9), a microRNA that when upregulated, helps maintain e&ndothelial function and characteristics. We have found that miR-9 is regulated by the long non-coding RNA ZFAS1 through targeted histone methylation in the heart. But given the disparate responses of retinal and cardiac endothelial cells to hyperglycemia, we wanted to investigate the upstream regulation of glucose-induced miR-9 inhibition in retinal endothelial cells. To this effect, we used human retinal and cardiac microvascular endothelial cells, treated with histone methylation inhibitors, DNA methylation inhibitors, or transfected with various siRNAs. We found that despite miR-9 being similarly affected by high glucose in retinal and cardiac endothelial cells, miR-9 is regulated through different epigenetic mechanisms in retinal and cardiac endothelial cells. Whereas cardiac endothelial cells relied on targeted histone methylation through ZFAS1, we showed that retinal endothelial cells relied more on DNA methylation, influenced by both ZFAS1 and the long non-coding RNA MALAT1, with ZFAS1 and MALAT1 influencing the expressions of one another. Our findings highlight the differences between regulatory mechanisms of similar processes across similar cell types, but point to miR-9 inhibition as a point of convergence between cardiac and retinal endothelial dysfunction may open the door for multi-organ targeted therapeutics.

## Introduction

Diabetes and its complications present major challenges to patients' quality of life, and roughly every 1 in 10 dollars in global healthcare expenditure is spent on diabetes-related care [1]. Strict blood glucose control is crucial in diabetes, and

**Data availability statement:** All relevant data are within the manuscript and its Supporting Information files.

**Funding:** This research was supported in part by grants from the Canadian Institutes of Health Research [grant number 173414] (SC), the Lawson Internal Research Fund from Lawson Health Research Institute (SC), and the Dean's Research Scholarship from Western University (EW). The funders had no input on the study design, data collection, data analysis, decision to publish, or preparation of the manuscript.

**Competing interests:** Authors have declared that no competing interests exist.

the lack thereof will, over time, result in the development of diabetic complications, such as diabetic retinopathy (DR) or diabetic cardiomyopathy (DCM) [2–4]. These complications negatively impact patients' qualities of life, with DR being the leading cause of vision impairments in working-aged adults, and DCM being a significant independent risk factor for heart failure in diabetic patients [2–4]. However, even with decent glycemic control, fluctuations in blood glucose largely means that onset of these complications can be effectively delayed but rarely fully avoided. Thus, a better understanding of the mechanistic development of these complications may provide additional benefits to people living with diabetes.

DR and DCM are both chronic microvascular complications of diabetes [2–4]. Microvascular complications are centered around pathologies of microvascular endothelial cells (MECs). MECs, and vascular endothelial cells in general, are highly susceptible to hyperglycemia-induced damage because they are in direct contact with glucose in the bloodstream and cannot dynamically regulate their glucose uptake [5–7]. The mechanisms of glucose-induced endothelial dysfunction have been well characterized; excessive glucose metabolism leads to the generation of reactive oxygen species (ROS) from the mitochondria, leading to the activation of pathways designed to shunt excess glucose into non-glycolytic, but otherwise detrimental, pathways in order to avoid excessive mitochondrial ROS generation [5–7]. MECs from different tissues have long been regarded as similar, and their responses to hyperglycemia have been assumed to be homogenous. MECs particular tissue origins have been used to model diabetic complications relating to other organs [8–10]. However, we have shown that MECs from different key target organs of diabetic complications respond uniquely to hyperglycemic insult [11]. It is thus important to consider MECs from different tissues independently.

Differing phenotypical responses to identical stimuli in cells with the same genetic background are mediated by epigenetic influences [4,12]. Epigenetics describes the regulation of gene expression that does not involve changes to the sequence of the genome [12]. Epigenetic mechanisms include histone modifications, DNA methylation, and regulation by non-coding RNAs (ncRNAs) [12,13]. Histone modifications involve the addition or removal of various ligands to the histone proteins and can increase or decrease gene expression [12,14]. DNA methylation involves adding or removing methyl groups to the DNA itself; methylation typically reduces gene expression [13,15]. Gene regulation by ncRNAs involve RNAs that are transcribed from the genome, but not translated into proteins [13,16]. They can interact with regulatory elements, protein complexes and mRNA transcripts to alter gene expression [13,16]. Epigenetic regulators interact and influence one another. For example, the long ncRNA (lncRNA) ZFAS1 binds to and guides the histone methylation complex, polycomb repressive complex 2 (PRC2) to mediate targeted suppression of specific loci in the genome [17], while lncRNA MALAT1 can bind to DNA methyltransferases (DNMTs) and guide targeted DNA methylation [18]. Alternatively, microRNAs (miRNAs) can specifically target the expression of epigenetic regulators such as histone modifiers, DNMTs or ncRNAs, to influence gene expression [19–22].

microRNA-9 (miR-9) is an conserved microRNA that is involved in a variety of diseases [23–25]. We have found miR-9 to be an important mediator of high

glucose-induced endothelial dysfunction in models of DR and DCM [13,17,22,26]. High glucose leads to suppression of miR-9 levels in both cardiac and retinal MECs, leading to endothelial dysregulation, specifically in the form of endothelial-to-mesenchymal transition (EndMT), where endothelial cells transdifferentiate into mesenchymal-like cells, losing endothelial attributes such as expression of platelet endothelial cell adhesion molecule (PECAM1), and adopting fibroblastic characteristics, such as increased expression of fibroblast-specific protein 1 (FSP-1/S100A4), contributing to basement membrane changes and fibrosis [17,22,26]. We have shown that endothelial-specific overexpression of miR-9 rescues the high glucose-induced phenotype in both types of MECs *in vitro* and *in vivo* [17,22,26].

Despite the downstream effects of miR-9 alteration being similar between cardiac and retinal MECs, transcriptomic analysis has shown that cardiac and retinal MECs have substantially different overall responses to hyperglycemia [11]. Furthermore, previous circular RNA analysis showed that expressions of several circular RNAs with complementarity to miR-9 were significantly altered in the heart, but not the retina of mice [27]. We have also explored the expression of lncRNA MALAT1, a lncRNA with complementarity for miR-9, in various target organs of diabetic complications and finding that its expression is transiently induced in the heart, but stably induced in the retina in response to diabetes [28]. Thus, apart from downregulation of miR-9 itself, it is not certain whether the regulation surrounding its downregulation would be similar across retinal and cardiac MECs. We have previously found that miR-9 in cardiac MECs may be regulated through the ZFAS1-PRC2 pathway [17]. We explored whether miR-9 in retinal MECs would have a similar mechanism of regulation. We hypothesized that miR-9 alteration may be a point of convergence in glucose-induced endothelial dysfunction between cardiac and retinal MECs, and that upstream regulation of miR-9 would be dissimilar between cardiac and retinal MECs.

## Methods

### Cell culture

Human retinal microvascular endothelial cells (HRMECs; Olaf Pharmaceuticals) and human cardiac microvascular endothelial cells (HCMECs; ScienceCell) were cultured with endothelial basal medium-2 (EBM-2; Lonza) supplemented with microvascular endothelial growth medium-2 (EGM-2; Lonza), with 10% v/v fetal bovine serum, and 100 µg/mL penicillin/streptomycin in a humidified culture hood at 37°C with 4% CO2. At 80% confluency, full growth medium (EBM-2 with EGM-2) was aspirated and replaced with serum-reduced medium for 24 hours. After 24 hours of serum-starvation, cells were treated to normal (NG, 5mM) or high (HG, 25mM) glucose concentrations for 48 hours. 48 hours was previously determined to be the optimal treatment duration to model early diabetes-induced endothelial changes in MECs [28]. Similarly, 25mM of glucose was determined to be the optimal concentration, where gene expression changes occur but cell viability is not affected (supplemental S1 Fig). L-glucose was used as osmotic controls.

### Transfection

Cells were transfected with siRNA against ZFAS1 or MALAT1. Several siRNA sequences were tested for each lncRNA, and the ones with the highest knockdown efficiencies were used. Transfection occurred prior to glucose treatments using lipofectamine 2000 (Invitrogen) in OPTI-MEM medium (Gibco) as previously described [22]. Scrambled oligonucleotides (SCR) were used as transfection controls. Following 6 hours of incubation in the transfection mixture, cells were recovered in full growth medium for 24 hours, after which cells were serum starved and treated to varying levels of glucose as mentioned above.

### Histone and DNA methylation inhibition

Pretreatment of cells with 3-Deazaneplanocin A (DZNep; Cayman Chemical) 1 hour prior to glucose treatment was used to inhibit histone methylation. Similarly, pretreatment with 5-Aza-2′-Deoxycytidine (5-Aza-dC; Cayman Chemical) was

used to inhibit DNA methylation prior to glucose treatment. Based on previous literature, 5 µM of DZNep or 5-Aza-dC was used for each well [29]. Cells were harvested 48 hours after treatment for analysis. Underlying data are available in supplementary information file.

## RNA analyses

For quantification of mRNAs and lncRNAs, total RNA was isolated using TRIzol reagent (Invitrogen) as previously described [22]. RNA concentrations were measured using a SpectraMax QuickDrop Spectrophotometer (Molecular Devices). 2 µg of RNA was used to synthesize cDNA using a High Capacity cDNA Reverse Transcription kit (Applied Biosystems) according to the manufacturer's instructions. qPCR was performed using TB green dye (Takara Bio) on a LightCycler 96 system (Roche Diagnostics). Gene expressions were quantified using the standard curve method and normalized to mRNA expressions of β-actin. See Table 1 for qPCR primers. Underlying data are available in supplementary information file.

For quantification of miR-9, total miRNA was isolated using the SanPrep Column microRNA Miniprep Kit (Biobasic) following the manufacturer's instructions. cDNA was synthesized using the aforementioned kit, but with specific primers, rather than random primers. qPCR was performed using TaqMan™ miR-9 assay (Ambion) in accordance with the manufacturer's instructions, on a LightCycler 96 system (Roche Diagnostics). miR-9 expression was quantified using the standard curve method and normalized to the expression of U6 snRNA. Underlying data are available in supplementary information file.

## Western Blot

Briefly, total protein was extracted from samples using RIPA buffer (Millipore) containing protease inhibitor (Roche). Protein concentration was assessed using Pierce bicinchoninic acid (BCA) assay kit (ThermoScientific). Thirty micrograms of protein was resolved via SDS-PAGE and transferred to a polyvinylidene difluoride membrane (PVDF; Bio-Rad). The PVDF membrane was blocked and incubated with primary antibodies overnight at 4°C, then with secondary antibody for

**Table 1. Primer sequences and thermocycler settings for qPCR.**

| Target | Primer | Sequence (5'–3') | Temperature profiles | |
|--------|--------|------------------|---------------------|---|
| ACTB | Forward<br>Reverse | CCTCTATGCCAACACAGTGC CATCGTACTCCTGCTTGCTG | Denaturation<br>Annealing<br>Extension<br>Signal | 95°C for 5s<br>55°C for 10s<br>72°C for 15s<br>84°C for 1s |
| MALAT1 | Forward<br>Reverse | TCTTAGAGGGTGGGCTTTTGTT<br>CTGCATCTAGGCCATCATACTG | Denaturation<br>Annealing<br>Extension<br>Signal | 95°C for 5s<br>55°C for 10s<br>72°C for 15s<br>80°C for 1s |
| PECAM1 | Forward<br>Reverse | AGACAACCCCACTGAAGACGTCG<br>CCTCTCCAGACTCCACCACCTTAC | Denaturation<br>Annealing<br>Extension<br>Signal | 95°C for 5s<br>55°C for 10s<br>72°C for 15s<br>82°C for 1s |
| S100A4 | Forward<br>Reverse | CAACAGGGACAACGAGG<br>CTGGGCTGCTTATCTGGG | Denaturation<br>Annealing<br>Extension<br>Signal | 95°C for 5s<br>55°C for 10s<br>72°C for 15s<br>84°C for 1s |
| ZFAS1 | Forward<br>Reverse | CAGCGGGTACAGAATGGA<br>TCAGGAGATCGAAGGTTGTAGA | Denaturation<br>Annealing<br>Extension<br>Signal | 95°C for 5s<br>55°C for 10s<br>72°C for 15s<br>82°C for 1s |

An initial denaturation phase was carried out at 95 °C for 2 minutes. 50 cycles were used for amplification

1 hour at room temperature (antibody information and dilutions in Table 2). Blots were visualized using Clarity™ Western ECL Substrate kit (Bio-Rad) and ChemiDoc™ MP Imaging System (Bio-Rad). Quantification was done using Image Lab software (Bio-Rad). Underlying data are available in supplementary information file.

## Promoter methylation assay

Genomic DNA was isolated using the DNeasy Blood and Tissue kit (Qiagen) according to the manufacturer's instructions. DNA concentration was measured using a SpactraMax QuickDrop Spectrophotometer (Molecular Devices). Bisulfite conversion was performed using EpiTect Plus DNA Bisulfite Kit (Qiagen) according to the manufacturer's. Methylation-specific primers [30] were used to quantify methylated and unmethylated miR-9 promoters (Table 3). Methylation statuses are presented as ratios of methylated:unmethylated (M:UM). Underlying data are available in supplementary information file.

## Chromatin immunoprecipitation (ChIP)

ChIP was performed using the EZ-Magna ChIP A/G kit (Sigma-Aldrich) according to the manufacturer's instructions. Briefly, cells were fixed in formaldehyde and harvested. Nuclear extraction was performed via ultrasonication. Immunoprecipitation was performed using anti-EZH2 or anti-DNMT1 antibodies (Abcam, ab191250, ab320817). Anti-IgG was used as a negative control and provided in the kit. The immunoprecipitated DNA was quantified using qPCR, and the enrichment of miR-9 promoter was calculated relative to IgG control using the $2^{-\Delta\Delta Ct}$ method. Underlying data are available in supplementary information file.

## Cell viability assay

The effect of miR-9 on cell viability was determined using the WST-1 assay (Roche) after 48 hours of high glucose treatment. Absorbances were measured at 450 nm, with a differential filter of 690 nm, using a ChroMate Microplate Reader(Awareness Technology). Underlying data are available in supplementary information file.

## Statistical analysis

Student's T-test and one-way ANOVA analyses were performed using Prism 10 (GraphPad) were appropriate. Tukey's honest significance test was used for pair-wise comparisons following ANOVA. The threshold of significance was set at $p \leq 0.05$.

**Table 2. Antibodies for western blot.**

| Antibody | Dilution |
|---|---|
| Rabbit antibody against β-actin (Abcam, ab8227) | 1:1000 |
| Rabbit antibody against PECAM1 (Abcam, ab281583) | 1:1000 |
| Mouse antibody against S100A4 (Proteintech, 66489–1-Ig) | 1:1000 |
| Goat anti-rabbit IgG-HRP (Santa Cruz, SC-2004) | 1:5000 |
| Goat anti-mouse IgG-HRP (Invitrogen, A28177) | 1:5000 |

**Table 3. Methylation specific PCR primers.**

| Target | Primer | | Sequence (5'–3') |
|---|---|---|---|
| *miR-9* promoter | Forward | Methylated | AGATTT**C**GTTTGGATGTTTTAGT**C** |
| | | Unmethylated | AGATTT**T**GTTTGGATGTTTTAGT**T** |
| | Reverse | Methylated | CAAAATACTTACC**GCG**CTTAA |
| | | Unmethylated | CAAAATACTTACC**ACA**CTTAAAA |

## Results

### Downregulation of ZFAS1 does not rescue miR-9 or downstream gene changes in HRMECs

We first confirmed that high glucose downregulated miR-9 in both cell types (Fig 1A), we also examined the effects of miR-9 abundance on cell viability, finding no significant changes (supplemental S2 Fig). We then confirmed similar levels of glucose-induced ZFAS1 induction across both cell types (Fig 1B). We have previously shown a ZFAS1-miR-9 axis in cardiac MECs, if a similar regulatory axis existed in retinal MECs, we would expect to see that knockdown of ZFAS1 under high glucose conditions would recover miR-9 levels and prevent endothelial dysfunction. We found, however, that siZFAS1 did not rescue miR-9 levels (Fig 1C). Furthermore, siZFAS1 also did not rescue markers of endothelial dysfunction, such as downregulation of the endothelial marker *PECAM1* and the induction of the mesenchymal marker *S100A4* in HRMECs at either the mRNA or the protein levels (Fig 2A–D; images of blots are in supplemental S3 Fig). Expressions of *PECAM1* and *S100A4* were corrected using a miR-9 mimic, either alone, or in conjunction with siZFAS1 (Fig 2E, F).

### Glucose-induced miR-9 suppression in HRMECs and HCMECs are mediated by different epigenetic mechanisms

As we have found that there was not a direct regulatory axis between ZFAS1 and miR-9 in retinal MECs as there is in cardiac MECs, we wondered if the mechanism of miR-9 suppression would also be different. In cardiac MECs, ZFAS1 targets PRC2 to the miR-9 locus to mediate inhibition of the locus via histone methylation [17]. Histone methylation inhibition using DZNep pre-treatment was able prevent high glucose-induced miR-9 inhibition in HCMECs (Fig 3A). Glucose-mediated miR-9 inhibition was not, however, rescued by DZNep in HRMECs (Fig 3A). ChIP also showed no increase in recruitment of PRC2 component to the miR-9 promoter locus following high glucose (Fig 3B), suggesting that histone methylation is not a major regulator of miR-9 in HRMECs as it is in HCMECs.

Having shown that histone methylation does not play a large role in miR-9 inhibition in HRMECs, we investigated whether DNA methylation plays a more significant role. DNMT1 ChIP showed significant enrichment of miR-9 promoter binding following high glucose treatment (Fig 3C). Furthermore, HRMECs showed a significant increase in miR-9 promoter M:UM ratio when treated to high glucose, while HCMECs showed a significant decrease (Fig 3D). Inhibition of DNA methylation using 5-Aza-dC pre-treatment prevented high glucose-induced miR-9 inhibition in HRMECs but not in HCMECs (Fig 3E). Expressions of endothelial functional marker *PECAM1* followed the same trend as miR-9 between the 2 MEC types (Fig 3F).

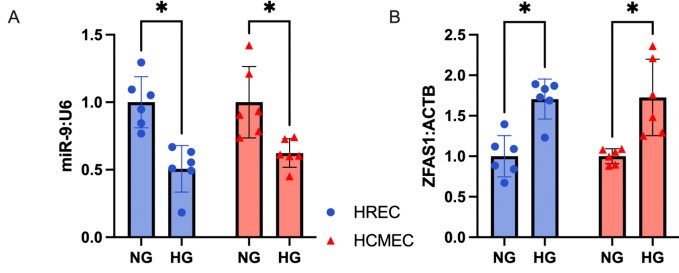
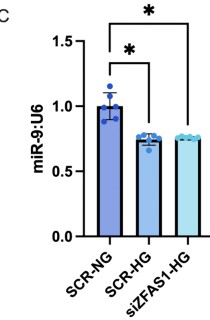

**Fig 1. Retinal MECs exhibit similar changes glucose-induced changes in miR-9 and ZFAS1 expression to cardiac MECs but lack a similar regulatory relation.** Expressions of A) miR-9 and **B)** ZFAS1 were altered in a similar direction and magnitude in retinal as they were in cardiac MECs in response to high glucose (HG; 25mM for 48 hours) treatment. **C)** Knocking down ZFAS1 in retinal MECs did not rescue the expression of miR-9. [n=6 for each experiment; RNA expressions presented as ratio to β-actin mRNA or U6 snRNA and normalized to normal glucose (NG; 5mM for 48 hours) or NG with scrambled siRNA control (SCR-NG) groups; all data presented as mean±SD; *=p<0.05].

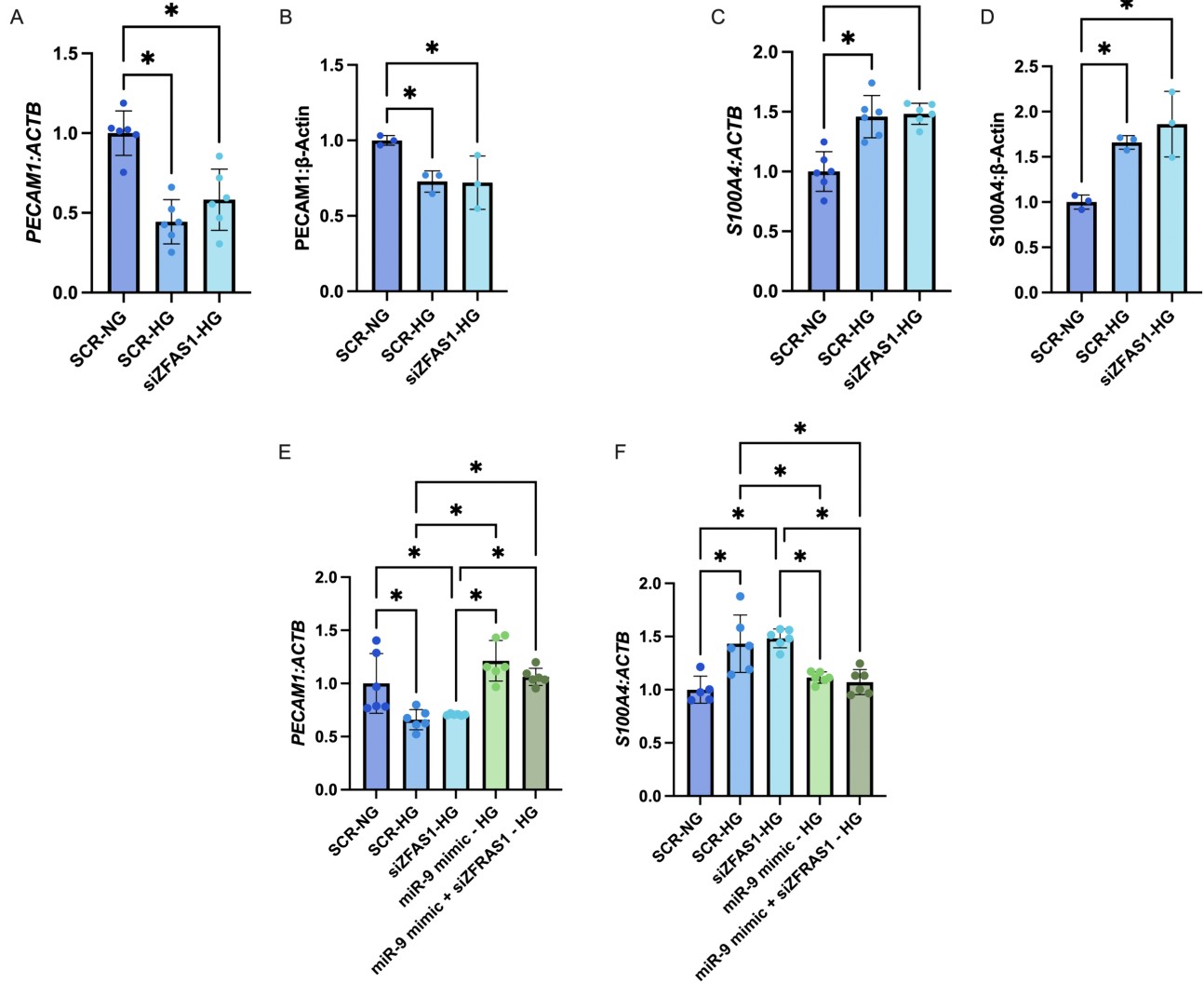

**Fig 2. knocking down ZFAS1 does not rescue markers of endothelial dysfunction in HRMECs.** Expressions of A) PECAM1 mRNA and B) PECAM1 protein were reduced following high glucose (HG; 25mM for 48 hours) treatment and not influenced by siZFAS1. Expressions of C) S100A4 mRNA and D) S100A4 protein were induced by HG treatment and was also not influenced by siZFAS1. miR-9 mimics were able to rescue the expressions of both E) PECAM1 and F) S100A4 either individually, or when co-transfected with siZFAS1. [n = 6 for RNA experiments, and n = 3 for protein experiments; RNA expressions presented as ratio to β-actin mRNA and normalized to normal glucose (NG; 5mM for 48 hours) with scrambled siRNA control (SCR-NG) groups; protein expressions presented as band intensity relative to β-actin; all data presented as mean ± SD; * = p < 0.05].

### ZFAS1 and MALAT1 differentially influence miR-9 promoter methylation in HRMECs

Having demonstrated that promoter methylation was more important in the regulation of miR-9 in HRMECs than histone methylation, we explored the potential relation between miR-9 promoter methylation, ZFAS1 and MALAT1. Knockdown of ZFAS1 under normal glucose conditions was associated with significantly increased M/UM ratio of the miR-9 promoter, compared to the transfection control group (Fig 4A). Promoter methylation was still higher in the siZFAS1 group cultured under high glucose, but the increase was not statistically significant compared to the normal glucose-treated group (Fig 4A). miR-9 expression was similarly reduced (Fig 4B). Knockdown of MALAT1, on the other hand, decreased the M/UM

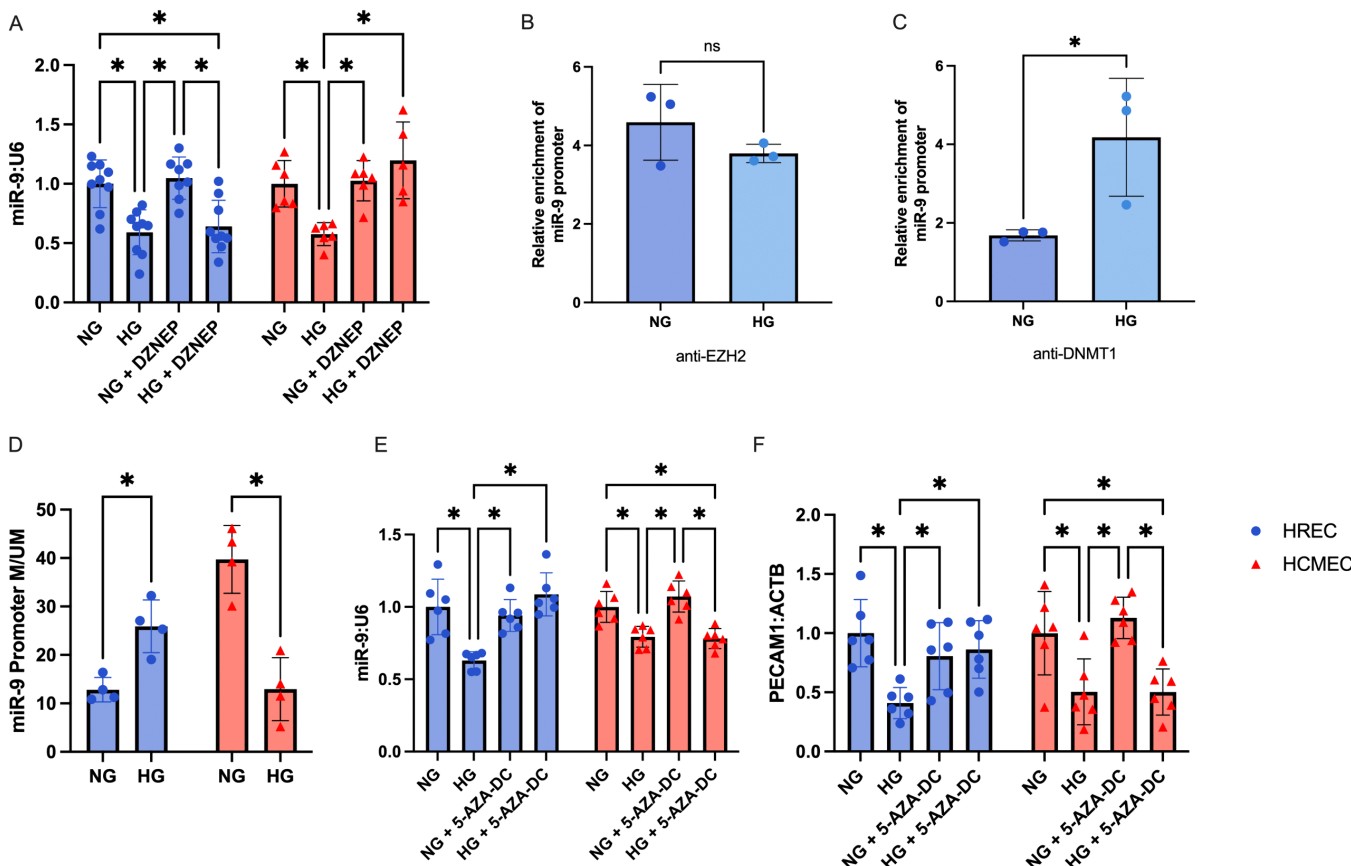

**Fig 3. Retinal and cardiac MECs are differentially influenced by histone vs DNA methylation. A)** Inhibition of histone methylation using DZNep rescued the high glucose (HG; 25mM for 48 hours)-induced suppression of miR-9 in cardiac MECs but not in retinal MECs. **B)** EZH2 ChIP showed no enrichment of EZH2 at the miR-9 promoter, while **C)** DNMT1 ChIP showed significant enhanced association of DNMT1 with the miR-9 promoter. **D)** Promoter methylation assay showed that HG caused hypermethylation of the miR-9 locus in retinal MECs but hypomethylation of the same locus in cardiac MECs. Inhibition of DNA methylation using 5-Aza-DC rescued HG-induced inhibition of E) miR-9 and **F)** PECAM1 expression in retinal MECs but not in cardiac MECs. [n = 9 for HRECs and n = 6 for HCMECs in the DZNEP experiment, n = 3 for ChIP, n = 4 for promoter methylation, n = 6 for both HRECs and HCMECs in the 5-AZA-DC experiment; RNA expressions presented as ratio to β-actin mRNA or U6 snRNA and normalized to normal glucose (NG; 5mM for 48 hours); ChIP data presented as fold enrichment relative to the IgG control; promoter methylation data presented as ratio of methylated to unmethylated; all data presented as mean ± SD; * = p < 0.05].

ratio of the miR-9 promoter, and was able to rescue miR-9 levels under high glucose conditions (Fig 4C, D). The expressions of downstream markers *PECAM1* and *S100A4* changed as expected, with respect to miR-9 expression (Fig 5A, B, D, E).

## ZFAS1 and MALAT1 show reciprocal regulation in HRMECs

While establishing opposing effects of ZFAS1 and MALAT1 on miR-9 promoter methylation in HRMECs, we wondered if there was any direct relation between ZFAS1 and MALAT1. Given that both lncRNAs are upregulated by high glucose treatment, we expected little influence by one lncRNA on the other. Surprisingly, knockdown of ZFAS1 prevented high glucose-induced MALAT1 induction (Fig 5C). The inverse was also observed, where knockdown of MALAT1 prevented high glucose-mediated induction of ZFAS1 (Fig 5F).

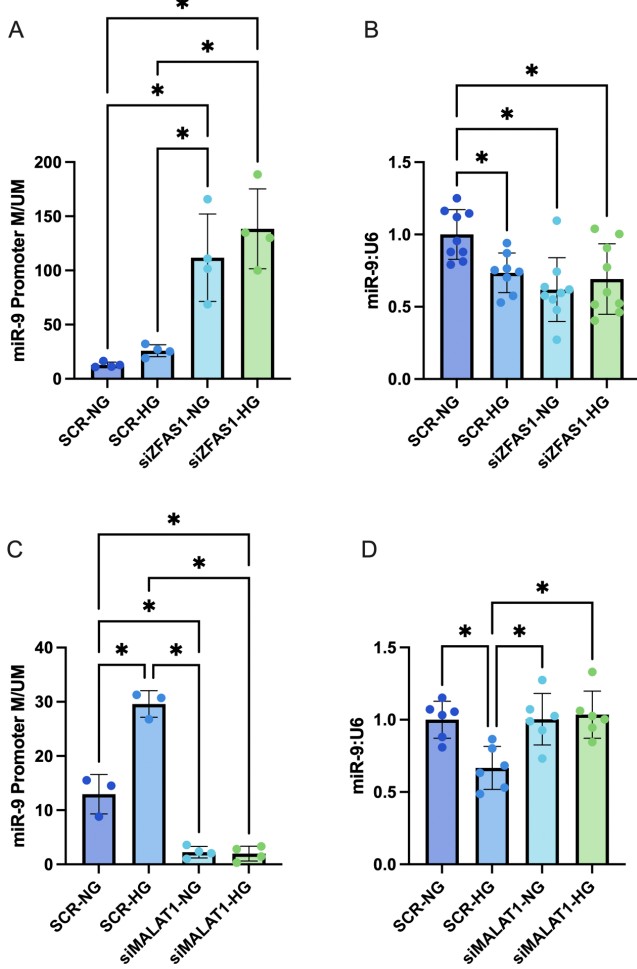

**Fig 4. lncRNAs ZFAS1 and MALAT1 both influence miR-9 promoter methylation in retinal MECs.** Inhibition of ZFAS1 caused A) hypermethylation of the miR-9 promoter locus, well above levels observed in the high glucose group transfected with scrambled siRNA controls (SCR-HG; 25mM for 48 hours). The hypermethylation in miR-9 promoter led to B) corresponding downregulation of miR-9. Inhibition of MALAT1 caused C) hypomethylation of the miR-9 promoter locus, and D) restoration of miR-9 expression under high glucose conditions. [n=4 for promoter methylation and n=9 for miR-9 expression in the siZFAS1 experiment, and n=3 for promoter methylation and n=6 for miR-9 expression in the siMALAT1 experiment; RNA expressions presented as ratio to U6 snRNA and normalized to normal glucose with scrambled siRNA control (SCR-NG; 5mM for 48 hours) group; promoter methylation data presented as ratio of methylated to unmethylated; all data presented as mean±SD; *=p<0.05].

## Discussion

High glucose-induced endothelial dysfunction is central to the development of chronic diabetic complications. We have separately shown that MECs from different organs respond distinctly to hyperglycemic insult, and that miR-9 may be a key regulator of EndMT across cardiac and retinal MECs [11, 17, 22, 31]. In this present study, we found that the upstream regulation of miR-9 in retinal MECs is different from cardiac MECs. We showed that histone methylation is a key regulator of miR-9 expression in cardiac MECs but not in retinal MECs. We further showed that miR-9 expression in retinal, but not cardiac MECs is strongly influenced by DNA methylation. We found that while ZFAS1 targeted PRC2 to the miR-9 locus in cardiac MECs, both MALAT1 and ZFAS1 had effects on miR-9 promoter methylation and downstream gene expression

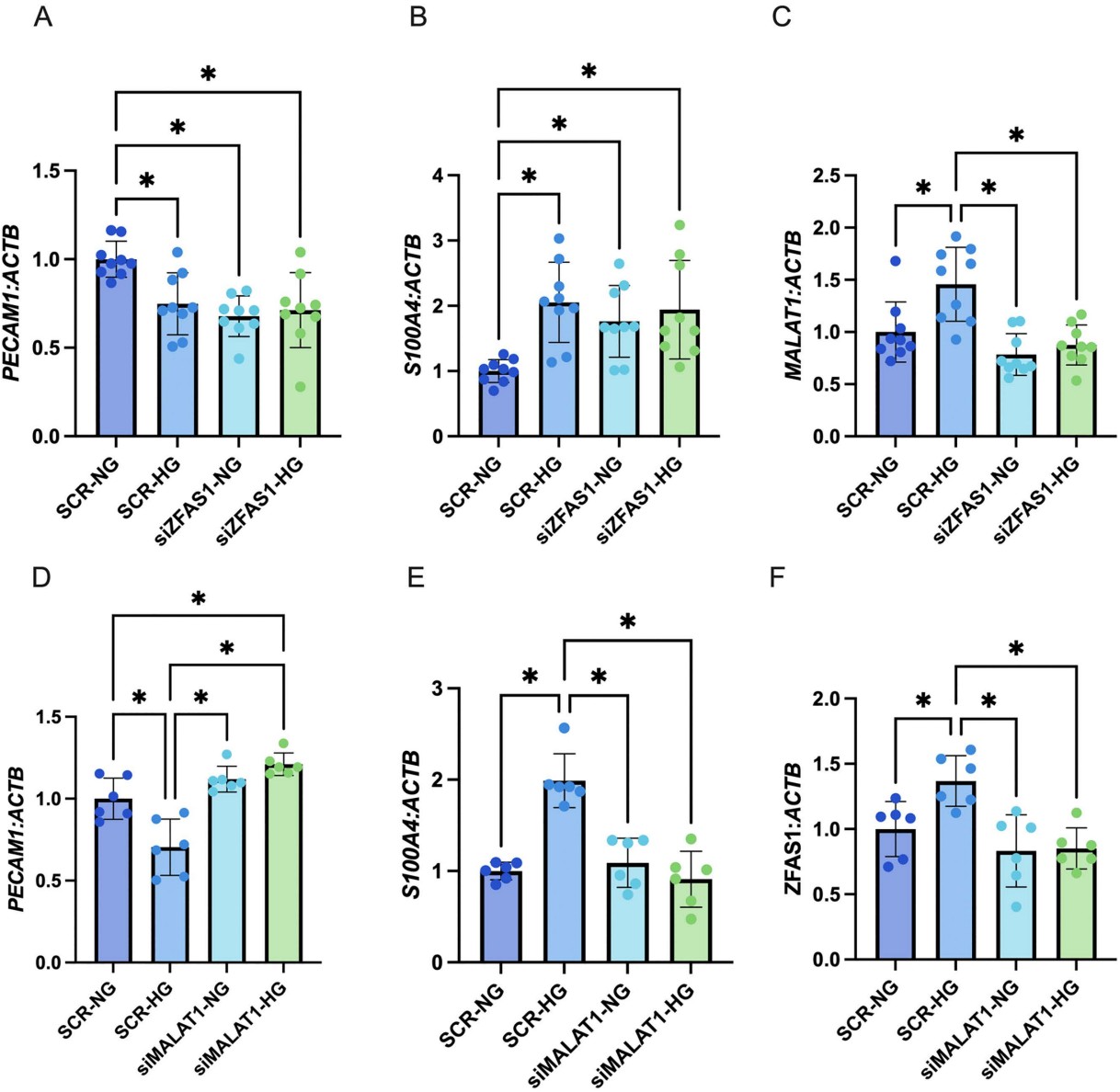

**Fig 5. Knockdown of lncRNAs ZFAS1 or MALAT1 leads to downstream changes related to their effects on miR-9 expression.** As a consequence of reducing miR-9, ZFAS1 knockdown caused high glucose-like changes in **A)** PECAM1 and **B)** S100A4 expressions in transfected groups treated to both normal glucose (NG; 5mM for 48 hours) and high glucose (HG; 25mM for 48 hours) groups. C) siZFAS1 also inhibited high glucose-induced MALAT1 induction. MALAT1 knockdown on the other hand, by rescuing miR-9 expression, rescued **D)** PECAM1 and **E)** S100A4 expressions. F) siMALAT1 also inhibited high glucose-induced ZFAS1 induction. [n = 9 in the siZFAS1 experiment, and n = 6 in the siMALAT1 experiment; RNA expressions presented as ratio to β-actin mRNA and normalized to NG with scrambled siRNA control (SCR) group; all data presented as mean ± SD; * = p < 0.05].

in retinal MECs. Finally, we demonstrated reciprocal influence of MALAT1 and ZFAS1 on one another in retinal MECs, indicating a more complex regulatory axis in retinal MECs compared to cardiac MECs.

Cardiac and retinal MECs serve similar but distinct functions within the body. Both form barriers between the blood and the organ, however, retinal MECs form a much more restrictive barrier known as the blood-retinal barrier, because

the retina is a much more metabolically sensitive tissue [32–34]. In addition to their different functions, cardiac and retinal MECs exist among different varieties of cells, leading to different microenvironments with different extracellular stimuli [35–38]. We have previously explored differential ncRNA expressions across target organs of diabetes, finding very little overlap between retinal and cardiac expressions of circRNAs [26, 27]. Hence the commonality of glucose-induced miR-9 inhibition and miR-9-related downstream changes across cardiac and retinal MECs was something that we wanted to further explore.

Although changes downstream of miR-9 inhibition, such as EndMT, were common across cardiac and retinal MECs, the mechanisms by which high glucose inhibits miR-9 are clearly distinct between the two MEC types. As demonstrated previously in cardiac MECs, ZFAS1 targets histone methylation machinery to the miR-9 locus, leading to suppression of miR-9 expression [17]. Inhibition of ZFAS1, which in cardiac MECs rescued miR-9 and downstream changes, did not do so in retinal MECs. The differential relation between ZFAS1 and miR-9 in cardiac and retinal MECs is further demonstrated by the lack of recovery in miR-9 levels in retinal MECs following global histone methylation inhibition. Together with the lack of enrichment of EZH2 at the miR-9 promoter regions following high glucose treatment, despite increased EZH2 expression in HRECs under high glucose, as we have previously found [39], this suggests that histone methylation is not a primary mechanism of regulation for miR-9 in retinal MECs. However, a lack of enrichment of PRC2 at the miR-9 locus does not mean that ZFAS1 does not bind to the PRC2 complex in retinal MECs as it does in cardiac MECs. ZFAS1 may still associate with PRC2 and attempt to target it to the miR-9 locus, but another epigenetic mechanism more strongly influences the miR-9 locus in retinal as compared to cardiac MECs. This could be an avenue for further investigation.

Given that glucose-induced miR-9 inhibition in retinal MECs appeared to not be driven by histone methylation through PRC2, exploration of DNA methylation was a natural follow-up. DNA methylation is straightforward; hypermethylation causes gene suppression, and hypomethylation is associated with increased gene expression [15]. We have previously shown that DNMT1 expression increases in HRECs in response to high glucose [39], however, expression may not correlate with activity. Here, we showed that DNA methylation inhibition rescued miR-9 levels in retinal MECs, where glucose caused significant hypermethylation of the miR-9 promoter. In cardiac EMCs however, DNA methylation inhibition showed little to no effect in cardiac MECs, where the miR-9 promoter became hypomethylated in response to high glucose. Hypomethylation of the miR-9 promoter in response to high glucose in cardiac MECs accompanied by downregulation of miR-9 may appear strange, given that hypomethylation is typically associated with increased expression. But studies have shown that DNA methylation and histone modifications, are incompatible at the same locus [40–43]. Evidence suggest a reciprocal antagonization between DNA and histone methylation, wherein DNA methylation inhibits the recruitment of PRC2 [40,43,44], and H3K27me3 inhibits DNA methylation [41,42]. Thus, the finding of miR-9 promoter hypomethylation in cardiac MECs in response to high glucose is in keeping with the increased targeting of PRC2-mediated histone methylation to the same locus. These findings suggest that cardiac MECs rely more on histone modifications to regulate miR-9 in response to high glucose, while retinal MECs rely more on DNA methylation.

The incompatibility between histone methylation and DNA methylation at the same locus helps untangle another aspect of high glucose-induced miR-9 inhibition in retinal MECs. MALAT1 influences DNA methylation at the miR-9 promoter in retinal MECs, and knockdown of MALAT1 significantly reduced the methylation of the miR-9 promoter and rescued the expression of miR-9. However, the significant increase in miR-9 promoter methylation following knockdown of ZFAS1 suggests the potential for a counterbalancing force by ZFAS1 at the miR-9 locus against DNA methylation. This is likely due to a similar targeting of PRC2 by ZFAS1 to the miR-9 promoter locus, as is the case in cardia MECs[17]. Knockdown of ZFAS1 likely leads to loss of PRC2 recruitment and eliminates the antagonistic effect of histone regulation on DNA methylation, leading to unopposed hypermethylation at the miR-9 promoter locus. Similar effects have been observed in PRC2-deficiency models, where the loss of PRC2 led to hypermethylation [45,46]. Thus, while MALAT1 and DNA methylation are the primary regulators of high glucose-induced miR-9 suppression in retinal MECs, ZFAS1 may be a key

modulator of the process. This appears to be a much more complex regulatory axis than in cardiac MECs, where MALAT1 is less relevant [28], and high glucose-related miR-9 suppression appears to be primarily mediated by ZFAS1.

Adding to the complexity of the regulation of miR-9 in response to high glucose in retinal MECs, is the observation that ZFAS1 and MALAT1 appear to reciprocally potentiate one another, with knockdown of either lncRNA limiting the glucose-induced upregulation of the other. Because we and others have shown that miR-9 directly targets MALAT1 via sequence complementarity, and we have presently shown that ZFAS1 knockdown suppresses miR-9, we expected MALAT1 levels to increase following siZFAS1 transfection under normal glucose conditions. The fact that it did not, and was even downregulated in under high glucose conditions following ZFAS1 knockdown, suggests more intricate mechanism of regulation within the MALAT1-ZFAS1-miR-9 axis. The inverse finding of high glucose-induced ZFAS1 upregulation being prevented by MALAT1 knockdown suggests that in retinal MECs, MALAT1 suppression will likely have wider effects, as it simultaneously rescues miR-9 and prevents ZFAS1 induction. This is less likely to be the case in cardiac MECs, where MALAT1 induction is transient and MALAT1 does not appear to play as significant of a role [28]. The mechanistic aspects of the reciprocal regulation between ZFAS1 and MALAT1 will require further investigation, but their main relevance in the context of the current study is their effects on miR-9, the main mediator of downstream EndMT. Glucose-induced miR-9 suppression appears to be one of the main outcomes of ZFAS1 and MALAT1 alterations, and is a key point of convergence between the retinal and cardiac MECs, making it a potential common target for DR and DCM. A schematic depiction of the MALAT1-ZFAS1-miR-9 axis is presented in Fig 6, to aid in understanding.

The translatability of the present study may be limited by the fact that all experiments were performed in vitro. This was a deliberate choice because it would be difficult to manipulate the various ncRNA expressions in an animal model. We have, however, already demonstrated the downstream effects of miR-9 downregulation and the beneficial effects of miR-9 overexpression in mouse models of DCM and DR previously [17,22]. The main aim of this study was to investigate the upstream regulation of miR-9 and the differential mechanisms behind its downregulation in response to high glucose.

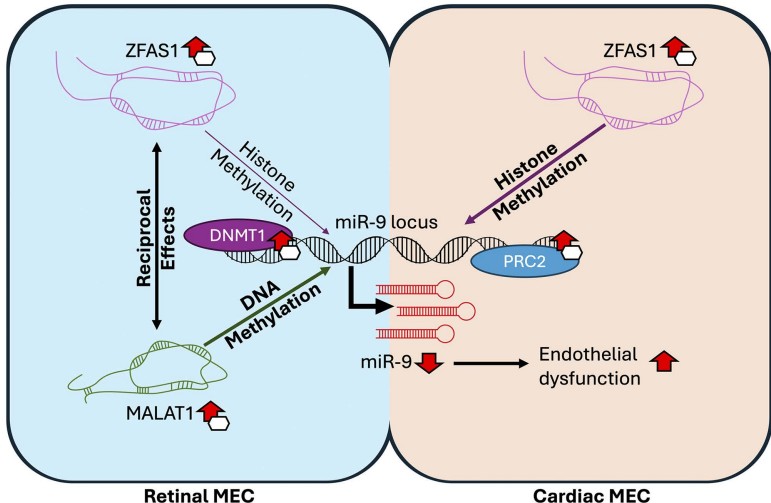

**Fig 6. Schematic overview of differential regulation of high glucose-induced miR-9 inhibition in retinal and cardiac MECs.** In cardiac MECs, high glucose induces ZFAS1 expression, which inhibits miR-9 expression via recruitment of PRC2 and induction of histone methylation. In retinal endothelial cells, high glucose induces the expressions of lncRNAs MALAT1 and ZFAS1. While ZFAS1 may still bind PRC2 and attempt to recruit it to the miR-9 locus, MALAT1 exerts a stronger influence by increasing DNMT1 recruitment to the same locus, resulting in miR-9 inhibition via DNA methylation. ZFAS1 remains a counter-balancing force to MALAT1, and there is reciprocal influence between ZFAS1 and MALAT1, painting a complicated ZFAS1-MALAT1-miR-9 axis in retinal MECs. Irrespective of upstream mechanisms, however, miR-9 suppression leads to the same outcome of high glucose-associated endothelial dysfunction in both MEC types.

As we have also previously demonstrated that differential endothelial responses to high glucose translates well between our cell models and our mouse model of diabetes, we believe that these in vitro findings are a good representation of the complex regulatory landscape existing in vivo.

In summary, we have shown that despite similar changes in miR-9 and downstream dysfunction in response to high glucose, retinal MECs have a different and slightly more complex upstream regulation of miR-9 expression compared to cardiac MECs. We found that high glucose-induced miR-9 inhibition in retinal MECs is mediated more so by promoter methylation, in contrast with cardiac MECs, where the process is more dependent on histone methylation. We have further shown that both ZFAS1 and MALAT1 influence miR-9 expression via influencing promoter methylation in retinal MECs. investigation will be required to elucidate the mechanism of reciprocal mediation between ZFAS1 and MALAT. In all, the study points to miR-9 as a convergence point in high glucose-induced dysfunction between retinal and cardiac MECs, making it a good candidate for potential RNA-based therapeutic approaches.

## Supporting information

**S1 Fig. Glucose concentration test.** Various concentrations of high glucose were tested to confirm the optimal glucose concentration for gene expression changes without severe glucotoxicity. S100A4 was used as a marker for downstream changes. High glucose (HG) at 20 and 25 mM showed significant differences compared to normal glucose (NG; 5mM). [n = 6; RNA expression presented as ratio to β-actin mRNA and normalized to the NG group; data presented as mean ± SD; * = $p < 0.05$].
(TIFF)

**S2 Fig. Effects of high glucose and miR-9 on cell viability.** High glucose (HG; 25mM) treatment did not reduce cell viability in retinal MECs. miR-9 overexpression or inhibition also do not significantly alter the OD450 readings, meaning they do not directly influence cell viability. [n = 10; SCR = scrambled siRNA control; NG = normal glucose (5mM); data presented as mean ± SD; * = $p < 0.05$].
(TIFF)

**S3 Fig. Western blot images.** Annotated images of A) blot 1 and B) blot 2, representing one single experiment, performed simultaneously. Raw images of C) blot 1 and D) blot 2 are also provided. [SCR = scrambled siRNA control; NG = normal glucose (5 mM); HG = high glucose (25mM)].
(TIFF)

## Author contributions

**Conceptualization:** Eric Wang.

**Data curation:** Eric Wang.

**Formal analysis:** Eric Wang.

**Funding acquisition:** Eric Wang, Subrata Chakrabarti.

**Investigation:** Eric Wang, Biao Feng, Shali Chen.

**Methodology:** Eric Wang, Biao Feng, Shali Chen, Subrata Chakrabarti.

**Supervision:** Biao Feng, Subrata Chakrabarti.

**Visualization:** Eric Wang.

**Writing – original draft:** Eric Wang.

**Writing – review & editing:** Biao Feng, Subrata Chakrabarti.

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
