## [Decision Letter · Decision Letter 0]

6 Jan 2026

PONE-D-25-67129Differential epigenetic regulation of glucose-induced alteration of miR-9 in retinal and cardiac endothelial cellsPLOS One

Dear Dr.  Chakrabarti,

Thank you for submitting your manuscript to PLOS ONE. After careful consideration, we feel that it has merit but does not fully meet PLOS ONE’s publication criteria as it currently stands. Therefore, we invite you to submit a revised version of the manuscript that addresses the points raised during the review process.

We look forward to receiving your revised manuscript.

Kind regards,

Filomena de Nigris, Ph.D.

Academic Editor

PLOS One

Journal Requirements:

4. Please note that funding information should not appear in any section or other areas of your manuscript. We will only publish funding information present in the Funding Statement section of the online submission form. Please remove any funding-related text from the manuscript.

5. We note that your Data Availability Statement is currently as follows: “All relevant data are within the manuscript and its Supporting Information files.”

6. We notice that your supplementary figures are uploaded with the file type 'Figure'. Please amend the file type to 'Supporting Information'. Please ensure that each Supporting Information file has a legend listed in the manuscript after the references list.

Reviewer's Responses to Questions

**Comments to the Author**

1. Is the manuscript technically sound, and do the data support the conclusions?

Reviewer #1: Yes

Reviewer #2: Partly

2. Has the statistical analysis been performed appropriately and rigorously? 

Reviewer #1: Yes

Reviewer #2: Yes

3. Have the authors made all data underlying the findings in their manuscript fully available?

Reviewer #1: Yes

Reviewer #2: Yes

4. Is the manuscript presented in an intelligible fashion and written in standard English?

Reviewer #1: Yes

Reviewer #2: Yes

5. Review Comments to the Author

Reviewer #1: The study “Differential epigenetic regulation of glucose-induced alteration of miR-9 in retinal and cardiac endothelial cells” is structured and easy to follow scientifically but can be improved with followings points.

Comments:

1. All data is from cultured cells. There is no animal or human tissue used. This limits the study. Better to add a sentence about this in the conclusion.

2. Some words are too strong like “firmly suggest”, “clearly demonstrate” for in vitro data only. Using soft language like suggests, support etc. will make reviewers more comfortable.

3. There is no time course data (only 48h) and only one glucose concentration is used.

4. MALAT 1 is said to influence DNA methylation, but there is no direct binding or recruitment shown. Similarily, ZFAS 1 is assumed to recruit PRC2 in retina because it does in heart, but not directly tested here. Clarify this.

5. Discussion section is very long and repetitive. Shorten it and focus on what is new and why it matters.

6. All the data is RNA based. There is no protein validation (western blot, IF, etc.). this make the study limited. Clarify this.

Reviewer #2: In the manuscript “Differential epigenetic regulation of glucose-induced alteration of miR-9 in retinal and cardiac endothelial cells” by Eric Zi Rui Wang et al., the authors investigate how high glucose conditions lead to the inhibition of microRNA-9 (miR-9) in cardiac and retinal endothelial cells. They show that, although miR-9 is similarly downregulated in both cell types, distinct epigenetic mechanisms are involved: histone methylation mediated by the long non-coding RNA ZFAS1 in cardiac endothelial cells, and DNA methylation regulated by both ZFAS1 and MALAT1 in retinal endothelial cells. These findings highlight miR-9 inhibition as a common pathway underlying endothelial dysfunction in different diabetic complications.

Below you will find my comments and some suggestions to improve the manuscript.

1. While knockdown of the lncRNAs ZFAS1 and MALAT1 is appropriately validated at the RNA level, it would strengthen the study to confirm the effects on downstream protein markers, such as PECAM1 and S100A4, by Western blot or immunofluorescence, to ensure that changes in RNA levels translate into protein-level effects and functional cellular outcomes.

2. The study focuses on molecular and epigenetic regulation of miR-9, but no functional assays were performed to directly assess endothelial cell behaviour. Including assays such as migration, tube formation, or proliferation/apoptosis would provide stronger evidence that the observed molecular changes translate into functional consequences for endothelial cells.

3. Overexpression of miR-9 with ZFAS1 or MALAT1 knockdown to test causal effects on endothelial phenotype should be performed to strengthen the results.

4. The study is largely in vitro and focuses on molecular mechanisms. The authors should acknowledge that physiological relevance in vivo remains to be tested.

5. In the Discussion some sentences are very long and complex (e.g., lines 258–261, 296–327); consider splitting them for readability.

6. Clarifying the MALAT1-ZFAS1-miR-9 reciprocal regulation in one concise sentence could improve comprehension.

7. Minor typographical error at page 13, line 110 (‘bay’ instead of ‘may’)

6. PLOS authors have the option to publish the peer review history of their article (what does this mean?). If published, this will include your full peer review and any attached files.

Reviewer #1: No

Reviewer #2: No

---

## [Author Response · Author response to Decision Letter 1]

6 Apr 2026

Reviewer #1: The study “Differential epigenetic regulation of glucose-induced alteration of miR-9 in retinal and cardiac endothelial cells” is structured and easy to follow scientifically but can be improved with followings points.

Comments:

1. All data is from cultured cells. There is no animal or human tissue used. This limits the study. Better to add a sentence about this in the conclusion.

We have included language to clarify the purely in vitro nature of this work.

2. Some words are too strong like “firmly suggest”, “clearly demonstrate” for in vitro data only. Using soft language like suggests, support etc. will make reviewers more comfortable.

We have adjusted the language.

3. There is no time course data (only 48h) and only one glucose concentration is used.

We have previously performed time point studies, and found that 48h was the optimal condition for studying early glucose-induced effects in both HCMECs and HRECs. We have included a glucose concentration test, showing that 25mM is the optimal concentration for our study purposes (supplementary figure S1).

4. MALAT 1 is said to influence DNA methylation, but there is no direct binding or recruitment shown. Similarly, ZFAS 1 is assumed to recruit PRC2 in retina because it does in heart, but not directly tested here. Clarify this.

We have performed ChIP assays to show increased recruitment of DNMT1 (and slightly decreased recruitment of EZH2) to the miR-9 locus following high glucose treatment (Figure 3). This demonstrates the importance of DNA methylation, rather than histone methylation, in miR-9 regulation in HRMECs. Together with the reduction of miR-9 promoter methylation following MALAT1 KD, this suggests that MALAT1 is involved in DNA methylation regulation. However, because lncRNAs have so many potential mechanisms of action, MALAT1 may or may not directly bind to DNMT1 for recruitment to the miR-9 locus. On the other hand, the role of ZFAS1 in PRC2 recruitment is postulated to occur in HRMECs in a similar way to HCMECs but remains unconfirmed. We have updated our language to clarify this distinction.

5. Discussion section is very long and repetitive. Shorten it and focus on what is new and why it matters.

We have modified the discussion.

6. All the data is RNA based. There is no protein validation (western blot, IF, etc.). this make the study limited. Clarify this.

We have previously validated downstream protein expression in our studies into miR-9 and its downstream effects. The present study focuses primarily on upstream interactions between miR-9 and other epigenetic regulators. We have however conducted a western blot to validate the effects of ZFAS1 KD on downstream targets in HRMECs (figure 2, supplementary figure S3), because it is different from what we have shown in the cardiac model.

Reviewer #2: In the manuscript “Differential epigenetic regulation of glucose-induced alteration of miR-9 in retinal and cardiac endothelial cells” by Eric Zi Rui Wang et al., the authors investigate how high glucose conditions lead to the inhibition of microRNA-9 (miR-9) in cardiac and retinal endothelial cells. They show that, although miR-9 is similarly downregulated in both cell types, distinct epigenetic mechanisms are involved: histone methylation mediated by the long non-coding RNA ZFAS1 in cardiac endothelial cells, and DNA methylation regulated by both ZFAS1 and MALAT1 in retinal endothelial cells. These findings highlight miR-9 inhibition as a common pathway underlying endothelial dysfunction in different diabetic complications.

Below you will find my comments and some suggestions to improve the manuscript.

Thank you for your comments, we have addressed the suggestions to the best of our ability.

1. While knockdown of the lncRNAs ZFAS1 and MALAT1 is appropriately validated at the RNA level, it would strengthen the study to confirm the effects on downstream protein markers, such as PECAM1 and S100A4, by Western blot or immunofluorescence, to ensure that changes in RNA levels translate into protein-level effects and functional cellular outcomes.

To confirm that lncRNAs exert their effects through regulation of miR-9 expression, we have validated the effects of ZFAS1 KD on downstream protein expression via Western blot (figure 2, supplementary figure S3).

2. The study focuses on molecular and epigenetic regulation of miR-9, but no functional assays were performed to directly assess endothelial cell behaviour. Including assays such as migration, tube formation, or proliferation/apoptosis would provide stronger evidence that the observed molecular changes translate into functional consequences for endothelial cells.

We have performed a proliferation/viability assay using WST-1. As we are examining the beginning stages of glucose-mediated changes, we are not expecting to see major functional changes at this time point. Similarly to what we have shown previously (doi.org/10.1038/s41598-018-24907-w), WST-1 showed minimal differences between NG and HG, and miR-9 also didn’t affect cell viability (supplementary figure S2). We expect that functional changes would be more apparent with a longer treatment period, but that is not within the scope of the current investigation. We have also clarified this within the text.

3. Overexpression of miR-9 with ZFAS1 or MALAT1 knockdown to test causal effects on endothelial phenotype should be performed to strengthen the results.

We have performed co-transfection with miR-9 mimic and ZFAS1 knockdown to illustrate the importance of miR-9 OE compared with the effect of ZFAS1 KD on downstream gene expressions (figure 2). Because we have shown that MALAT1 knockdown increases miR-9 expression, we did not perform miR-9 OE with MALAT1 KD.

4. The study is largely in vitro and focuses on molecular mechanisms. The authors should acknowledge that physiological relevance in vivo remains to be tested.

We have previously demonstrated the in vivo effects/relevance of high glucose-induced mir-9 downregulation in both DCM and DR. Here, we are focused on highlighting the differential upstream regulation of miR-9 between cardiac and retinal endothelial cells. We have updated our language to reflect the purely in vitro nature of this study.

5. In the Discussion some sentences are very long and complex (e.g., lines 258–261, 296–327); consider splitting them for readability.

We have adjusted the language for some longer sentences.

6. Clarifying the MALAT1-ZFAS1-miR-9 reciprocal regulation in one concise sentence could improve comprehension.

We have attempted to clarify the regulatory axis. It is also more clearly explained in the schematic in Figure 6.

7. Minor typographical error at page 13, line 110 (‘bay’ instead of ‘may’)

We have corrected the typo.

---

## [Decision Letter · Decision Letter 1]

22 Apr 2026

PONE-D-25-67129R1Differential epigenetic regulation of glucose-induced alteration of miR-9 in retinal and cardiac endothelial cellsPLOS One

Dear Dr. Chakrabarti,

Thank you for submitting your manuscript to PLOS ONE. After careful consideration, we feel that it has merit but does not fully meet PLOS ONE’s publication criteria as it currently stands. Therefore, we invite you to submit a revised version of the manuscript that addresses the points raised during the review process. Please submit your revised manuscript by Jun 06 2026 11:59PM. If you will need more time than this to complete your revisions, please reply to this message or contact the journal office at plosone@plos.org.  Please include the following items when submitting your revised manuscript:

We look forward to receiving your revised manuscript.

Kind regards,

Filomena de Nigris, Ph.D.

Academic Editor

PLOS One

Journal Requirements:

Reviewers' comments:

Reviewer's Responses to Questions

**Comments to the Author**

1. If the authors have adequately addressed your comments raised in a previous round of review and you feel that this manuscript is now acceptable for publication, you may indicate that here to bypass the “Comments to the Author” section, enter your conflict of interest statement in the “Confidential to Editor” section, and submit your "Accept" recommendation.

Reviewer #1: All comments have been addressed

Reviewer #2: All comments have been addressed

2. Is the manuscript technically sound, and do the data support the conclusions?

Reviewer #1: Yes

Reviewer #2: Yes

3. Has the statistical analysis been performed appropriately and rigorously? 

Reviewer #1: Yes

Reviewer #2: Yes

4. Have the authors made all data underlying the findings in their manuscript fully available?

Reviewer #1: Yes

Reviewer #2: Yes

5. Is the manuscript presented in an intelligible fashion and written in standard English?

Reviewer #1: Yes

Reviewer #2: Yes

6. Review Comments to the Author

Reviewer #1: Thank you to the authors for the revision. The manuscript has improved significantly, especially with the addition of the ChIP data and protein validation. The study is now much clearer and better supported. I only have a few suggestions.

1. Please clearly state the number of biological replicates (n) for each experiment in the figure legends or method section. It would also help to clarify whether these were independent experiments.

2. For the ChIP experiments, please briefly clarify how enrichment was calculated (e.g., relative to IgG or input). A short clarification would improve transparency.

3. Since DNMT1 recruitment is important in the retinal mechanism, it would be helpful to mention whether DNMT1 expression itself changes under high glucose (if data are available).

4. A small amount of editing in the discussion to simplify the explanation of the MALAT1–ZFAS1–miR-9 interaction would improve readability.

Reviewer #2: (No Response)

7. PLOS authors have the option to publish the peer review history of their article (what does this mean?). If published, this will include your full peer review and any attached files.

Reviewer #1: No

Reviewer #2: No

---

## [Author Response · Author response to Decision Letter 2]

24 Apr 2026

Reviewer #1: Thank you to the authors for the revision. The manuscript has improved significantly, especially with the addition of the ChIP data and protein validation. The study is now much clearer and better supported. I only have a few suggestions.

1. Please clearly state the number of biological replicates (n) for each experiment in the figure legends or method section. It would also help to clarify whether these were independent experiments.

We have added n values to each figure legend

2. For the ChIP experiments, please briefly clarify how enrichment was calculated (e.g., relative to IgG or input). A short clarification would improve transparency.

We had briefly mentioned of the enrichment calculation relative to IgG in the figure legend, but we have now added more information directly to the methods section.

3. Since DNMT1 recruitment is important in the retinal mechanism, it would be helpful to mention whether DNMT1 expression itself changes under high glucose (if data are available).

We have not quantified DNMT1 expression levels in the current study. However, we have previously (doi: 10.1167/iovs.62.3.20) that RNA expressions of DNMTs and PRC2 components are similarly upregulated under high glucose conditions. We have added comments to this effect in the discussion. [lines 338-339, and lines 348-350]

4. A small amount of editing in the discussion to simplify the explanation of the MALAT1–ZFAS1–miR-9 interaction would improve readability.

We have simplified the discussion surrounding the MALAT1-ZFAS1-miR-9 axis. [lines 379-427]

---

## [Editor Report · Decision Letter 2]

28 Apr 2026

Differential epigenetic regulation of glucose-induced alteration of miR-9 in retinal and cardiac endothelial cells

PONE-D-25-67129R2

Dear Dr. Chakrabarti,

We’re pleased to inform you that your manuscript has been judged scientifically suitable for publication and will be formally accepted for publication once it meets all outstanding technical requirements.

Kind regards,

Filomena de Nigris, Ph.D.

Academic Editor

PLOS One

---

## [Editor Report · Acceptance letter]

PONE-D-25-67129R2

PLOS One

Dear Dr. Chakrabarti,

I'm pleased to inform you that your manuscript has been deemed suitable for publication in PLOS One. Congratulations! Your manuscript is now being handed over to our production team.

Kind regards,

on behalf of

Prof. Filomena de Nigris

Academic Editor

PLOS One